# What Are The Core Symptoms of Antenatal Depression? A Study Using Patient Health Questionnaire-9 among Japanese Pregnant Women in the First Trimester

**DOI:** 10.3390/healthcare11101494

**Published:** 2023-05-20

**Authors:** Toshinori Kitamura, Yuriko Usui, Mikiyo Wakamatsu, Mariko Minatani, Ayako Hada

**Affiliations:** 1Kitamura Institute of Mental Health Tokyo, Tokyo 104-0061, Japan; yusui@g.ecc.u-tokyo.ac.jp (Y.U.); hada@institute-of-mental-health.jp (A.H.); 2Kitamura KOKORO Clinic Mental Health, Tokyo 151-3306, Japan; 3T. and F. Kitamura Foundation for Studies and Skill Advancement in Mental Health, Tokyo 151-0063, Japan; 4Department of Psychiatry, Graduate School of Medicine, Nagoya University, Nagoya 464-8601, Japan; 5Department of Midwifery and Women’s Health, Division of Health Sciences and Nursing, Graduate School of Medicine, The University of Tokyo, Tokyo 113-0033, Japan; 6Department of Reproductive Health Care Nursing, School of Health Sciences, Faculty of Medicine, Kagoshima University, Kagoshima 890-8544, Japan; mikiwaka@health.nop.kagoshima-u.ac.jp; 7Life Value Creation Unit, NTT DATA Institute of Management Consulting, Inc., Tokyo 102-0093, Japan; minatanim@nttdata-strategy.com; 8Department of Community Mental Health & Law, National Institute of Mental Health, National Center of Neurology and Psychiatry, Tokyo 187-8551, Japan

**Keywords:** antenatal depression, emesis, diagnostic entity

## Abstract

**Background:** Depression is frequently seen among pregnant women. This is called antenatal depression (AND). **Aim:** Our aim was to identify clusters of AND and its core symptoms. **Methods:** The Patient Health Questionnaire-9 (PHQ-9), Pregnancy-Unique Quantification of Emesis and Nausea (PUQE-24), and Nausea and Vomiting of Pregnancy Quality of Life Questionnaire (NVP-QOL) were distributed to 382 pregnant women with a gestational age of 10 to 13 weeks who were attending antenatal clinics. The two PHQ-9 subscale scores were entered into a 2-step cluster analysis. The PHQ-9 items’ capacity to identify AND were examined in terms of the area under curve (AUC) of a receiver operating characteristic (ROC) analysis. The selected symptom items were examined for their diagnostic capability in terms of the graded response model (GRM) in the item response theory (IRT) analysis. **Results:** Three clusters emerged. Cluster 3 scored highly in the scores of the two PHQ-9 subscales and the two emesis scales. In the ROC, five items showed an AUC > 0.80. The GRM identified four items with high information: ‘loss of interest’, ‘depressed mood’, ‘self-esteem’, and ‘poor concentration’. **Conclusions:** The core symptoms of antenatal depression were four non-somatic symptoms; particularly, ‘depressed mood’ and ‘loss of interest’. AND did not exist alone, but was accompanied by nausea and vomiting. Hence, we propose a new category: emesis–depression complex among pregnant women.

## 1. Introduction

Depression during pregnancy is a major health issue. The pooled prevalence estimate of antenatal depression is 21%, and that of major antenatal depression is 15% [1]. Usually, studies use a standard diagnostic criterion such as the Diagnostic and Statistical Manual of Mental Disorders (DSM). Such definitions of depressive episodes (depressive disorder) date back to Feighner et al.’s [2] St. Louis criteria, in which depression was identified when five or more items were identified out of the following nine (at least one of the first two was indispensable): depressed mood, lack of interest, appetite problems, sleep problems, psychomotor symptoms, fatigue, reduced self-esteem, reduced concentration, and suicidality. These criteria have stood the test of history, remaining in the Research Diagnostic Criteria [3] and DSM-III to the current DSM-V. However, the validity of the above set of diagnostic items has long been debatable. Even in 1984, Endicott [4] cast doubt on the application of the DSM-III criterion of a major depressive episode (MDE) among patients with cancer. Intuitively, we presume that somatic symptoms such as insomnia, anorexia, fatigue, and retardation among cancer patients are likely somatic (physiological) symptoms (particularly under chemotherapy) and should not be treated as MDE indicators. This was supported by a study using an item response theory analysis [5]. A similar argument was made by Cavanaugh [6], who reviewed the diagnostic validity of the DSM-IV MDE criteria and suggested modifications to the descriptions in the texts. In the same vein, some researchers have avoided somatic symptoms when creating an assessment inventory (e.g., the Hospital Anxiety and Depression Scale (HADS) [7] and Beck Depression Inventory (BDI) [8]). Expectant women often experience emesis/hyperemesis gravidarum (HG). Emesis/HG is one of the uncomfortable symptoms experienced by pregnant women. HG affects approximately 1.1% of pregnancies [9]. It is doubtful to apply the “five-out-of-nine rule” to depressive states experienced by pregnant women without modification. For example, Matthey and Ross-Hamid [10] interviewed 118 pregnant women, asking about symptoms of depression and anxiety together with an additional enquiry about whether the endorsement of a symptom was due to their physical changes in pregnancy. More than a half of them who met the criteria for MDE answered that a sufficient number of their symptoms were due to the normal physical changes of their pregnancy, so that they no longer met the criteria for a diagnosis of MDE. A shortcoming of Matthey and Ross-Hamid’s [10] study was that they relied on the participants’ opinion about whether symptoms experienced were due to the ‘normal’ process of pregnancy or mental illness. Thus, we are still uncertain about which symptoms are free from the effects of physical changes and which are ‘real’ (core) symptoms of MDE during pregnancy.

Several previous studies have reported an association between emesis/HG and depression or other mental illnesses [11,12,13,14,15,16,17], and it is clear that emesis/HG and depression coexist [18,19,20,21]. Even though a decrease or an increase in appetite and weight loss or gain due to emesis/HG and pregnancy-related sleep disturbances are also present in normal pregnant women, the diagnostic criteria for depression in non-pregnant women have automatically been applied to pregnant women. As a result, common discomforts of pregnancy may also be diagnosed as depression. This may have led to an inflation in the prevalence of antenatal depression (AND).

A possible means to disentangle this complicated symptomatology and diagnostic categorization is a cluster analysis accompanied by item response theory (IRT). A cluster analysis is a statistical method for the classification of similar objects into groups based on the observed values of several variables for each individual. Classification has always played an essential role in science [22]. A cluster analysis is useful to identify subgroups. It may be feasible to expect that expectant women with AND constitute a category that is discrete from those who are normal or with only nausea and vomiting (pure emesis/HG). There may appear to be a cluster of women who have somatic symptoms such as nausea and vomiting, but without the core symptoms of AND. We presume that there are symptoms that distinguish the AND cluster from both emesis/HG and healthy women. Such symptoms should be regarded as core indicators (symptoms) of AND. Therefore, after identifying a cluster of pregnant women with AND, we planned to examine the contribution of *each* diagnostic item (indicator/symptom) so that we could select the best set of symptom items to diagnose AND. Here, we used IRT. When measuring depression, the greater the amount of information obtained from the item score, the more useful the item score for distinguishing between people in terms of the severity of depression [23,24,25]. We expected that some items that were indicators of MDE among non-perinatal women might lose their diagnostic specificity when applied to expectant women.

## 2. Methods

### 2.1. Study Procedures and Participants

This was a secondary analysis of our previously reported epidemiological dataset [26,27,28,29]. Approximately 1500 pregnant women in the period of 10 to 13 weeks’ gestational age were recruited at the antenatal clinic of one general hospital and five private clinics located in Tokyo, Chiba, Ibaraki, and Kagoshima Prefectures in Japan. Exclusion criteria included those (a) who were not fluent in Japanese, (b) who were aged under 20, (c) who had eating disorders, (d) who had symptoms of vaginal bleeding or abdominal pain, (e) who had subchorionic hematomas, or (f) who had experiences of recurrent miscarriages. We administered a set of questionnaires on 2 occasions, 1 week apart. The total sample consisted of 382 pregnant women (approximately 25% of those who were solicited) who participated in this study. Participation was voluntary and anonymity was assured. The mean (SD) age of the participants was 31.9 (4.9) years old and that of their partners was 33.5 (5.5) years old. Most of them were married (94%); 44% of the participants were nulliparas and 55% were multiparas. The recruitment of this study was conducted from January 2017 to May 2019.

### 2.2. Measurements

For major depressive episode (MDE) and depression symptoms, we used the Patient Health Questionnaire-9 (PHQ-9) [30], a self-reported nine-item depression scale based on the criteria of MDE of DSM-Ⅳ. Each item checks the frequency of depressive symptoms over the previous two weeks. Each item is calculated with a 4-point Likert scale from 0 to 3. The psychometric properties’ validity and reliability of the Japanese version of PHQ-9 were reported [31,32]. We already reported a result of a factor analysis of the PHQ-9 items among current expectant women that yielded two subscales, somatic and non-somatic [26]. The somatic subscale included the items ‘sleep’, ‘fatigue’, and ‘appetite’, whereas the non-somatic subscale included the items ‘loss of interest’, ‘depressed mood’, ‘self-esteem’, ‘poor concentration’, ‘psychomotor’, and ‘suicidality’. In addition, we used the indicator of MDE according to Spitzer et al.’s [30] criteria. This was a dichotomous measure of MDE based on the DSM-IV diagnostic criteria. Here, each DSM item was rated as ‘present’ if it was answered more than ‘more than half the days’ and, as the MDE criterion, ‘present’ items at least were required to diagnose MDE. If the number of ‘present’ items was less than 5, the case was judged as non-MDE.

The nausea and vomiting symptoms were rated by two scales, the Pregnancy-Unique Quantification of Emesis and Nausea (PUQE-24) [33] and the Nausea and Vomiting of Pregnancy Quality of Life Questionnaire (NVP-QOL) [34]. PUQE-24 is a scale that rates (a) nausea (the length of nausea in hours for the last 24 h), (b) vomiting (number of vomiting episodes in the last 24 h), and (c) retching (the number of retching episodes in the last 24 h), each with a 5-point scale. Hada et. al. [27] validated its psychometric properties in Japanese. NVP-QOL is a questionnaire with 30 items to evaluate nausea and vomiting of pregnancy (NVP) and related QOL in the previous week with a 7-point Likert scale (1 = none of the time to 7 = all of the time). Its 1-factor structure was identified in Japan [29]. In addition, we created an ad hoc item, ‘nausea therapy’, which rated outpatient visits or inpatient admissions for emesis/HG (1 = neither, 2 = outpatient visit, and 3 = inpatient admission).

For social disability, we used the Sheehan Disability Scale (SDS (Copyright 1983–2020, Sheehan, D.V. All rights reserved. May be reproduced only with the permission of Dr David V. Sheehan, copyright holder. For permission, contact davidsheehan@gmail.com)), which contains three items that are simple questions to assess disability covering (a) work and schoolwork, (b) social and leisure activities, and (c) family life and home responsibilities [35]. The validation of the Japanese version was reported [28].

For insomnia, we used the Insomnia Severity Index (ISI) [36,37], which is a questionnaire to assess the nature, severity, and impact of insomnia by asking about sleep for the last two weeks and consists of seven items with a 5-point Likert-type scale (from 0 = no problem to 4 = very severe problem). The 2-factor structure of the Japanese version of the ISI consists of Severity (severity of sleep onset, sleep maintenance, early morning awakening problems, and sleep satisfaction) and Impact (interference in sleep difficulties with daytime functioning, noticeability of sleep problems by others, and distress caused by the sleep difficulties) [38].

### 2.3. Data Analysis

First, we calculated the mean, SD, skewness, and kurtosis for the scores of somatic and non-somatic subscales of PHQ-9. Four missing cases in PHQ-9 were treated with a listwise deletion. Then, a 2-step cluster analysis was conducted to classify cases depending on the two PHQ-9 subscale scores. One-way ANOVAs with Tukey post hoc comparisons were conducted to compare sociodemographic and other characteristics. After the identification of clusters, the screening performance of *each* item of PHQ-9 was estimated as an area under curve (AUC) by a receiver operating characteristic (ROC) curve [39].

The identification of the PHQ-9 items that had a substantial capacity to detect AND cases led to IRT analyses of such items. The graded response model (GRM) [40,41] was fitted with item response theory (IRT) to confirm the item discrimination parameters and item information parameters for latent AND traits. Item information curves (IICs) were also graphed. IICs show the amount of information about an ability level that an item score provides at each point on the ability scale [24]. As we will discuss later, non-somatic symptoms were more reflective of AND than somatic symptoms. Item 9, ‘suicidality’, was included as a non-somatic factor; however, its AUC was low (probably due to its very low basic rate). Although ‘suicidality’ is an extremely important symptom in depression, it appears only in particularly severe cases. ‘Suicidality’ should be treated with another independent domain as an important task in perinatal mental health. Therefore, we excluded item 9, ‘suicidality’, as well as somatic items from IRT to identify the core symptoms of AND.

Before fitting the GRM, the IRT assumptions were examined. The 1-factor model of the non-somatic domain was fitted to the data by a confirmatory factor analysis (CFA). An exploratory factor analysis (EFA) was performed if the CFA showed a poor fit [42,43]. The 1-factor model was assured if the greatest factor accounted more than 20% of the variance as well as more than 4 of the ratio of the first and second factors’ eigenvalues (PROPMIS Standard 2.0). Local independence was examined by residual correlations from the 1-factor CFA where any item pairs with >0.2 residual correlation were flagged [42,44]. Monotonicity was evaluated using scalability coefficients (coef_*Hi*) (>0.3) [44,45] using the Mokken scale.

Analyses were performed by SPSS^®^ ver.28 (IBM. Inc., Armonk, NY, USA). In addition, we used R Studio V.07.2-576. EFAs were conducted using psych version 2.1.3 [46], residual correlations were calculated using lavaan version 0.6-11 [47], monotonicity was examined using mokken version 3.0.6 [48], and the GRM was examined via ltm version 1.2-0’ [49] packages in R.

### 2.4. Ethical Consideration

This study was approved by the Institutional Review Board (IRB) of the Kitamura Institute of Mental Health Tokyo (No. 2015052301) and Kagoshima University (No. 170247).

## 3. Results

Both somatic and non-somatic subscale scores showed skewness < 2.0 and kurtosis < 4.0, thus suggesting a normal distribution (Table 1). The 2-step cluster analysis using the two PHQ-9 subscale scores revealed 3 clusters. Its silhouette measure of cluster cohesion was >0.5. Clusters 1, 2, and 3 consisted of 135, 144, and 98 cases, respectively (Table 2). Both somatic and non-somatic subscale scores were statistically significantly higher in the order of Clusters 3, 2, and 1. The prevalence of MDE (according to the algorithm of Spitzer et al. [30]) was 0%, 0.7%, and 46% among Clusters 1, 2, and 3, respectively. A noteworthy fact was that the cases of Cluster 3 were more likely to reflect non-somatic symptoms than Cluster 2 (see Figure 1). Hence, Cluster 3 was characterized by a predominance of non-somatic items as well as the presence of MDE compared with the other two clusters.

The scores of PUQE-24 and NVP-QOL were again statistically significantly higher in the order of Clusters 3, 2, and 1 (Table 2). Although not statistically significant, the nausea therapy score showed a similar trend across the three clusters. Cluster 2 was, therefore, characterized by nausea and vomiting and PHQ-9 somatic subscale scores compared with Cluster 1. Therefore, we considered that those women in Cluster 1 were normal, those in Cluster 2 had nausea and vomiting, and those in Cluster 3 had AND together with nausea and vomiting. The last group may have indicated the existence of nausea/vomiting and depression. We failed to identify a cluster characterized only by depression (with an absence of nausea and vomiting).

The scores of the SDS and ISI subscales were again statistically significantly higher in the order of Clusters 3, 2, and 1 (Table 2). There was no significant difference among the cases belonging to all clusters in terms of own age and parity. Partner’s age was statistically higher among Cluster 2, but its difference was slight.

The screening performances of *each* item of PHQ-9 for Cluster 3 (against Clusters 1 and 2 combined) were estimated (Table 3) in terms of the AUC of the ROC. ‘Loss of interest’, ‘depressed mood’, ‘self-esteem’, and ‘poor concentration’ showed an AUC > 0.8. These all belonged to the non-somatic subscale. Of the somatic subscale items, only ‘fatigue’ had an AUC > 0.8. However, fatigue is a very common symptom for pregnant women. The mean score of ‘fatigue’ was the highest among the women belonging to Clusters 1 or 2 combined (Table 3). Therefore, we identified the above four non-somatic symptoms (together with one more non-somatic item, ‘psychomotor’) as a latent trait of core symptoms of AND, proceeding to an item analysis using IRT.

The 1-factor model consisting of the above five items did not show an acceptable fit with the data in the CFA (CFI = 0.929; RMSEA = 0.155) (Appendix A). An EFA was conducted for the examination of unidimensionality. The scree test suggested a 1-factor model. Residual correlations for each item ranged between −0.069 and 0.088 (<0.2), and coefficient *H* ranged between 0.399 and 0.584 (>0.3) (Appendix A). The IRT assumptions became assured, and the GRM was fitted to the data. The item discrimination parameter was higher in the order of ‘loss of interest’ (2.722), ‘depressed mood’ (2.714), ‘self-esteem’ (2.365), ‘poor concentration’ (2.171), and ‘psychomotor’(1.315). Item information was higher in the order of ‘depressed mood’ (6.37), ‘loss of interest’ (6.34), ‘self-esteem’ (5.18), ‘poor concentration’ (4.70), and ‘psychomotor’ (2.52) (Table 4). The item information curve of ‘psychomotor’ showed a substantially lower curve than the other items (Figure 2). Thus, ‘psychomotor’ did not have a sufficient ability to measure the AND latent trait. Hence, we identified the other four items as core symptoms of AND. As expected, ‘loss of interest’ and ‘depressed mood’ were the two most reflecting symptoms of AND.

## 4. Discussion

In this study, a cluster analysis derived three clusters. Cluster 3 consisted of about a quarter of the sample of women, and was the highest in both the PHQ-9 somatic and non-somatic subscale scores. When the DSM MDE diagnostic criterion (the 5-out-of-9 rule) was strictly applied, the prevalence of MDE was 45% among the women of Cluster 3 compared with virtually naught of those of Clusters 1 and 2. The Cluster 3 women also scored the highest in social disability (SDS) and insomnia (ISI). The SDS scores in this group were higher than those of individuals without mental disorders [50,51]. These findings suggest that women in this cluster experienced daily life difficulty, thus this cluster was that of AND. The fact that only half of the women in this cluster met the MDE criterion indicated that these women included cases of not only MDE but also minor and intermittent depression [19]. The boundary of AND identified by the cluster analysis was wider than that defined by DSM-V. It is also noteworthy that the Cluster 3 women scored the highest in terms of nausea and vomiting. As there appeared to be no cluster characterized only by depressive symptoms without signs of nausea and vomiting, we considered that Cluster 3 women reflected a condition of both AND and emesis. Hence, we named this Cluster ‘emesis–depression complex’. In 1993, Kitamura et al. [19] interviewed expectant women, of whom 16% were diagnosed as suffering from depression and reported that such women were more likely to complain about nausea.

The women of Cluster 2 scored higher than Cluster 1 women in terms of the scores of the somatic and non-somatic subscales, but only 2% of them met the MDE criterion. These women scored higher than Cluster 1 women in terms of nausea and vomiting measures (PUQE-24 and NVP-QOL). Hence, we named Cluster 2 ‘nausea and vomiting’. They scored lower than Cluster 3, but higher than Cluster 1 in terms of the PHQ-9 subscales. This was in line with past investigations that women with emesis/HG had higher depression scores [14,16,18,20]. These past studies were, however, based on instruments of depression severity measures and did not use diagnosis-specific measures (such as PHQ-9) or structured diagnostic interviews, thus we are unaware of the proportion of AND women in these studies.

An issue of clinical importance is the distinction between emesis and HG. Our Cluster 3 may have included cases of HG. We were uncertain about this issue because we did not measure ketonuria or electrolytes. However, a differentiation between emesis and HG is unclear, even among obstetricians. For example, Koot et al. [52], in a systematic review, examined all the RCTs of HG (defined by researchers) and checked the definition of HG. From 34 reports, components of a definition were vomiting (34; 100%), nausea (30; 88%), weight loss (9; 26%), dehydration (7; 21%), inability to tolerate oral food/water intake (4; 12%), ketonuria or acidosis (19; 56%), electrolyte disturbances (5; 15%), and a need for hospitalization (17; 50%). This suggests a substantial variation in the definition of HG. Therefore, it is obvious that obstetricians have different diagnostic categories for emesis and HG and there are, to the best of our knowledge, no empirical studies (such as cluster analyses and taxometrics) about the distinction of emesis and HG. Future studies are recommended to measure psychological as well as somatic symptoms simultaneously and identify diagnostic categories.

The AUC showed a screening performance capacity of each depression item. The AND women differentiated from normal and nausea and vomiting women (and normal women) by ‘loss of interest’, ‘depressed mood’, ‘reduced self-esteem’, and ‘poor concentration’. This was endorsed by IRT analyses, demonstrating that these four items were core symptoms of AND. Among those symptoms, ‘loss of interest’ and ‘depressed mood’ were the most reflecting symptoms of AND. This was in line with Huey et al. [53] who identified core symptoms of MDE among palliative care patients.

Pregnant women with AND may be erroneously treated for emesis or HG in perinatal settings, but it may be necessary to carefully assess them and to aggressively provide psychological care for *depressive* symptoms in addition to treatments for nausea and vomiting. It may be useful to make therapeutic plans based on the possible causal variables described in the vast amount of literature on correlates of AND. Kitamura et al. [54] listed as correlates of AND (1) obstetric factors (first pregnancy, first childbirth, and past history of artificial abortion), (2) early experiences such as a loss of father, (3) personality, including Eysenck’s high neuroticism, (4) negative attitudes towards the current pregnancy, (5) poor accommodation such as non-detached housing and expected crowdedness after childbirth, and (6) a lack of social support such as a low level of partner intimacy. Of particular note is that Salomonsson [55] proposed a psychodynamic treatment for pregnancy-related mental disorders. Evans et al.’s [56] systematic review suggested that women with antenatal anxiety perceived a benefit from peer support and individual discussions of their situation.

Our results suggested a new set of symptoms for defining depression. Although the current 5-out-of-9 convention dates back to half a century ago, there have been debates on its validity. For example, Angst and Dobler-Mikola [57], in their epidemiological study on a young adult population, noted that the gender ratio of depression was higher among women when applying DSM-III or RDC, but equal between the two genders when relying only on the existence of dysphoric mood (joyless, depressive, sick of life, loss of interest, loss of efficiency, and feelings of inferiority) and work disability. This was because women with a dysphoric mood reported much greater numbers of depression-related symptoms (such as appetite, weight, sleep, and psychomotor symptoms) than men with dysphoria. Accordingly, Angst and Dobler-Mikola [58] proposed that the number of required depression-related symptoms should be different between the two genders; 3 and 5 for men and women, respectively. Unfortunately, they did not perform IRT analyses on their data. However, their studies suggested that we should be cautious about the definition of the boundary of *clinical* depression. When reporting their monumental paper, Feighner et al. [2] wrote that “This communication is meant to provide common ground for different research groups so that diagnostic definitions can be amended constructively as further studies are completed”. They further added that “These criteria are not intended as final for any illness”. Our data indicated that ‘depressed mood’ and ‘loss of interest’ were the two core symptoms of depression that potently differentiated cases of depression from others, even when colored by nausea and vomiting.

This study had several limitations. First, we only studied pregnant women in the first trimester. Although the worst time for emesis/HG is in the first trimester, pregnant women with emesis also exist in the second and third trimesters. As many pregnant women with emesis/HG through the second and third trimesters have mental health problems, future studies should identify the core symptoms of depression throughout the pregnancy period. Second, the low response rates limited the generalizability of the results. Pregnant women with more severe emesis/HG or depression may not have been able to answer the questions. Third, our sole reliance on questionnaire data should be amended by interview data. This, however, requires a structured interview specially designed for a pregnant woman population. We also became aware of shortcomings of PHQ-9. Its somatic items lump diametrically opposed content into one item, such as insomnia/hypersomnia, decreased/increased appetite, and psychomotor agitation/retardation. If each direction was asked separately, the results might have been different. Of course, as our sample was all Japanese women, we should be sensitive to cultural or linguistic differences. Cross-cultural studies should be performed colleting populations from multiple countries. This was, however, beyond the scope of the current study, and awaits future international studies.

## 5. Conclusions

Our study showed that the core symptoms of AND were four non-somatic symptoms, particularly ‘depressed mood’ and ‘loss of interest’. It also demonstrated that AND did not exist alone, but was accompanied by nausea and vomiting in the first trimester. Hence, we propose a new category among pregnant women: emesis–depression complex.

## Figures and Tables

**Figure 1 healthcare-11-01494-f001:**
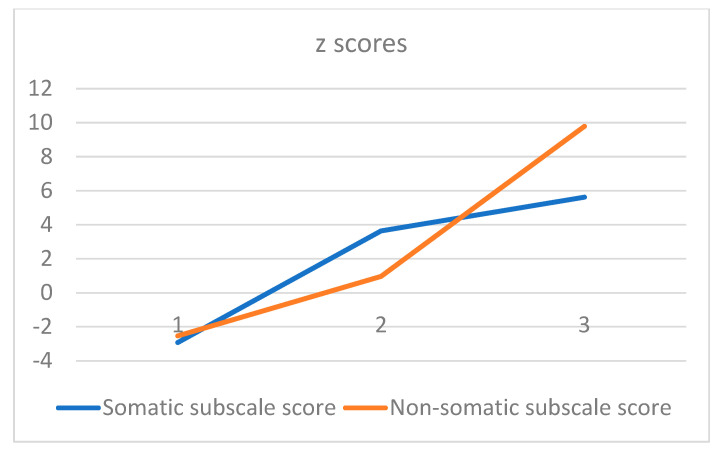
Somatic subscale and non-somatic subscale scores (z score) for each cluster.

**Figure 2 healthcare-11-01494-f002:**
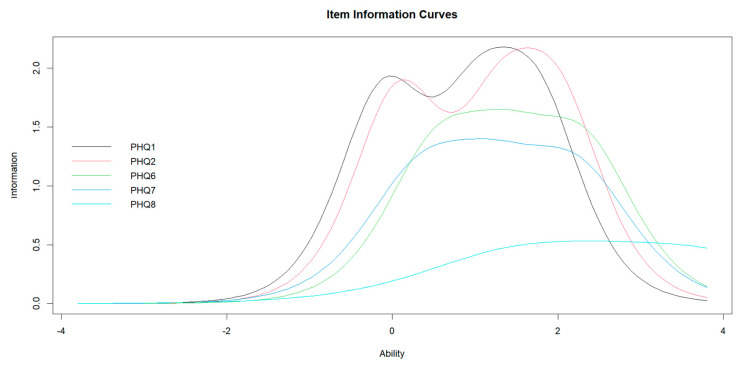
Item information curves of GRM for *non-somatic* items. PHQ: Patient Health Questionnaire.

**Table 1 healthcare-11-01494-t001:** Mean, SD, skewness, and kurtosis for the subscales of PHQ-9 (*n* = 378).

Subscales	Mean	SD	Skewness	Kurtosis
Somatic	7.63	2.55	0.80	−0.99
Non-somatic	8.79	3.18	1.40	1.60

**Table 2 healthcare-11-01494-t002:** Cluster characteristics.

Cluster	Cluster 1 (*n* = 135)	Cluster 2 (*n* = 144)	Cluster 3 (*n* = 98)	*F*	Post Hoc Comparison
Somatic subscale (z score)	−2.92	3.64	5.63	382.9 ***	1 < 2 < 3
Non-somatic subscale (z score)	−2.53	0.96	9.79	458.3 ***	1 < 2 < 3
MDE, *n* (%)	0 (0.0)	1 (0.7)	45 (45.9)	140.6 ***	–
PUQE-24 (z score)	2.51	4.13	5.23	30.8 ***	1 < 2 < 3
NVP-QOL (z score)	73.59	105.10	137.76	88.8 ***	1 < 2 < 3
Nausea therapy (z score)	−0.20	−0.16	0.03	1.7 *^NS^*	–
SDS (z score)	1.99	6.90	12.98	88.1 ***	1 < 2 < 3
ISI severity (z score)	3.02	6.53	8.46	63.5 ***	1 < 2 < 3
ISI impact (z score)	1.19	3.95	4.68	38.0 ***	1 < 2 < 3
Own age	25.48	24.75	25.84	0.8 *^NS^*	
Partner’s age	27.33	28.88	28.17	3.7 *	1, 3 < 2
Nulliparae, *n* (%)	59 (44.0)	66 (46.5)	41 (41.8)	χ^2^ = 0.52 *^NS^*	–
Multiparae, *n* (%)	75 (56.0)	76 (53.5)	57 (58.2)		–

* *p* < 0.05; *** *p* < 0.01; *^NS^*, not significant; ISI: Insomnia Severity Index; MDE: major depressive episode; NVP-QOL: Nausea and Vomiting of Pregnancy Quality of Life Questionnaire; PUQE-24: Pregnancy-Unique Quantification of Emesis and Nausea; SDS: Sheehan Disability Scale.

**Table 3 healthcare-11-01494-t003:** Mean SD and AUC of ROC (Cluster 3 vs. the others).

No.	Item	Clusters 1 + 2	Cluster 3	AUC
1	Loss of interest	1.42 (0.58)	2.84 (0.92)	0.883
2	Depressed mood	1.31 (0.49)	2.47 (0.88)	0.858
3	Sleep	2.24 (1.06)	3.16 (0.89)	0.735
4	Fatigue	2.25 (0.86)	3.32 (0.73)	0.810
5	Appetite	2.36 (1.00)	3.34 (0.81)	0.761
6	Self-esteem	1.16 (0.41)	2.39 (0.95)	0.858
7	Poor concentration	1.23 (0.49)	2.64 (0.88)	0.896
8	Psychomotor	1.10 (0.35)	1.78 (0.96)	0.701
9	Suicidality	1.01 (0.09)	1.12 (0.36)	0.553

**Table 4 healthcare-11-01494-t004:** Threshold parameters, discrimination parameters, and item information of GRM.

PHQ-9		b1	b2	b3	a	Information
1	Loss of interest	−0.093	1.060	1.711	2.722	6.34
2	Depressed mood	0.085	1.345	1.974	2.714	6.37
6	Self-esteem	0.572	1.387	2.285	2.365	5.18
7	Poor concentration	0.379	1.215	2.200	2.171	4.70
8	Psychomotor	1.468	2.436	3.724	1.315	2.52

PHQ-9: Patient Health Questionnaire-9; b: item threshold parameter; a: item discrimination parameter.

## Data Availability

The dataset analyzed and used in this study may be available upon reasonable request to the first author.

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
