# Peer review of "What Are The Core Symptoms of Antenatal Depression? A Study Using Patient Health Questionnaire-9 among Japanese Pregnant Women in the First Trimester"

_healthcare, 2023, doi:10.3390/healthcare11101494_

Round 1

Reviewer 1 Report

Thank you for inviting me to review this manuscript.

Overall it is well written and adds the body of knowledge in the area of perinatal mental health management and diagnosis. I note that this is a secondary analysis and is thus considered exploratory in nature. I also note low response rates and small numbers limiting generalisability. 

I have attached my feedback for your consideration.

There are some minor English language issues that Im confident will be picked up during normal editing procedures.

Author Response

Abstract: It is unclear what the aim of the study is. The use of a structured reporting guideline is recommended to report the abstract in a standardised way.

We set up Background, aim, Method, Results, and Conclusion. Actually, it seems that our submitted manuscript was rewrote in the present version by the editorial office. We wonder which is the Journal’s style.

Introduction line 46: ‘It is even 1984 that Endicott’ – recommend rewording

We changed its expression as follows;

Even in 1984, Endicott [4] cast doubt on application of DSM-III criterion of Major Depressive Episode (MDE) among patients with cancer. Intuitively, we presume that somatic symptoms such as insomnia, anorexia, fatigue, and retardation among cancer patients are likely somatic (physiological) symptoms (particularly under chemotherapy) and should not be treated as MDE indicators.

Line 56: Its is unclear what HG is – please define as its an important factor used throughout this paper.

HG stands for hyperemesis gravidarum.

Line 57: ‘HG is effecting…..’, incorrect use of English, consider change to HG affects approximately 1%....

We changed its expression as follows;

HG affects approximately 1.1% of pregnancies (Einarson et al., 2013).

The use of the term ‘normal women’ suggests women outside of this category are abnormal. Please consider using another term.

We presume that there are symptoms that distinguish the AND cluster from both emesis/HG and healthy women.

Methods: This seems to be an exploratory study (secondary analysis)? If this is the case consider outlining as such in the title.

We agree with the reviewer’s suggestion and changed the title as ‘What are core symptoms of antenatal depression: A study using the Patient Health Questionnaire-9 among Japanese pregnant women in the first trimester’.

25% response rate T1 and only 129 T2 (with some missing data). This doesn’t seem to be mentioned again only Time 1. Please clarify.

Line 105: Its mentioned here that measurements were take at two timepoints, but this does not seem to mentioned again??? Why was this second time-point undertaken, what is the relevance to the current study reported?

The reviewer is correct. Although we followed the participants 1 week later, these data are irrelevant to the current report. Therefore, related sentences were deleted.

2.2 Measurements Line 114: The PHQ-9 is used. Is this a translated version and how reliable/valid is the tool in a Japanese population of pregnant women? – I appreciate you signpost to one of your earlier papers but please provide some information here.

Of course, it was the Japanese translated version and we added references of its psychometric.

Line 123: In addition we used the indicator of MDE according to Spitzer et al . This is a dichotomous measure…. Please expand this to be more clear. What is the criteria and what are the two categories??

We added the following sentences.

Here, each DSM item was rated as ‘present’ if it was answered more than ‘mor than half the days’ and, as the MDE criterion, at least ‘present’ items were required to diagnose MDE. If the number of ‘present’ items are less than 5, the cases was judged as non-MDE.

Line 125: Its still unclear to me why you’ve chosen to include nausea and vomiting as an outcome. Maybe its relevance needs to be outlined more in the background section.

We think we spent whole paragraphs in Introduction to explain why we were interested in depression and emesis among expectant women.

Line 126:  Measurement tools. Consider including some information regarding the validity and reliability of the tools specific to the translated versions enabling the reader to make a judgement on the robustness of the chosen tools

We prepared necessary refences to the measurement tools regarding its psychometric properties.

Data analysis: Line 163: Item 9 suicidality – AUC was low….was this reflective of low numbers, its unclear why this would be???

We added possible explanation as follows;

The item 9 ‘suicidality’ was included in Non-Somatic factor, however, its AUC was low (probably due to its very low basic rate).

Line 169: …’were to be examined’ –, consider changing to ….were examined.

We changed expression as ‘Before fitting GRM, IRT assumptions were examined’.

Line 170: Would EFA not be conducted first and then CFA???

We follow usual steps of GRM. Here, a set of items is presumed as a 1-factor model. This is checked by CFA but if CFA does not prove a 1-factor solution, researchers should go back to EFA.

Line 186 and Table 1: I don’t think you need table 1 – this data can be outlined in the text to use less space – Im not sure it adds much. Or consider including as supplementary information.

We think that means (SD) and other indices are to be presented for Reader’s sake.

Results There is no table regarding characteristics of the respondents. Please include a table. How many women responded to experiencing nausea and vomiting or suicidality for example??

Most of the variables were continuous and therefore we presented means and SDs.

Results: are for 377 women so Im assuming you used the first time point that data was collected. What is the relevance to the second time point outlined in methods section.

The reviewer is correct. Although we followed the participants 1 week later, these data are irrelevant to the current report. Therefore, related sentences were deleted.

Line 208: We failed to identify a cluster characterised only by depression (in absence of N+V). Does this mean N+V is always present??

We are talking about the characteristics of the identified clusters but cases.

Discussion Line 260: You name cluster 3 ‘emesis-depression complex’. And cite Kitamura as reporting depressed women more likely to complain of nausea. Whay did all the women in the current study experience N&V??

Please remember that we are describing the characteristics of the three clusters. It is not that all the cases belonging to the same cluster have the same scores of depression, nausea, and vomiting. The reason why depression is combined with emesis remains to be further studies. Our manuscript highlights the link between the two that have thus far considered as separate. Depression is seen by psychiatrists whilst emesis by obstetricians and midwives.

Line 291 and Line 320: Reference 51 and 2 – include name of author before citation

We rewrite is accordingly.

Line 298 In a move towards more ‘woman-centred’ language, consider avoiding the term ‘delivery’ and consider changing to the term ‘birth’. Women give birth to babies – Pizzas are delivered.

We changed expression accordingly.

Line 336 You note the important limitations to your study and note shortcomings of the PHQ-9 the primary outcome measure of this study making it a major limitation. Can you please discuss further and make recommendations for further research in the future. You don’t appear to identify any strengths for you study – please include.

We have already noted shortcomings of the PHQ-9 (as below) and suggested further steps to take.

We also became aware of shortcomings of the PHQ-9. Its somatic items lump diametrically opposed content into one item such as insomnia/hypersomnia, decreased/increased appetite, and psychomotor agitation/retardation. If each direction was asked separately, the results might be different.

As to strength we repeatedly wrote about combination of depression and emesis during pregnancy.

Conclusion You make a proposal for a new category : emesis-depression complex in pregnant women. This might be considered a bold preposal based on the limitations of the study. As an exploritory study using secondary analysis, further research would surely be recommended to confirm your findings and using diagnostic interview in addition.

Thank you. We will.

Table 2: Consider including abbreviations on the bottom of the table.

All the abbreviations were explained in Table 2.

References: References 26, 27, 28, 29 are all the authors own work but do appear to be cited appropriately within the manuscript to support the psychometric properties of included outcome measures. References 40, 41 and 42 all relate to PROMIS data bank. Its unclear why these references are used. Consider if another citation might be more appropriate to support methodology

As indicated, references 26, 27, 28, 29 are all our own works. We are unaware of any other references to support psychometrics. We cannot understand why references to PROMIS is inappropriate here. 

Reviewer 2 Report

This study collected information from pregnant women in Japan. After a series of statistical analyses, the authors proposed a new category to identify or diagnose, and describe these women’s depressive symptoms; that is, "the emesis-depression complex” for pregnant women in Japan. Included below are several comments and hope the authors can provide their clarifications.

When the authors set up and justified their study, they argued that the past applications and implications of the Diagnostic and Statistical Manual of Mental Disorders (DSM) have been debated in diagnosing major depression. This is fine so long as the authors specify if this was for the general population or for the sub-population such as pregnant women. Was the debate pertaining to the Japanese population or the populations in the West? Would this make any difference? Moreover, did the authors argue or imply that the Patient Health Questionnaire-9 (PHQ-9) works better than the DSM? If so how and why? Or simply because the instrument was included in their study? This type of claim must be well-guarded!

Only a small portion of the pregnant women participated in the study. Any possible selection biases?

Please report or indicate the goodness-of-fit of the GRM model(s). Do these models fit the data?

A thorough line edition is warranted.

Author Response

When the authors set up and justified their study, they argued that the past applications and implications of the Diagnostic and Statistical Manual of Mental Disorders (DSM) have been debated in diagnosing major depression. This is fine so long as the authors specify if this was for the general population or for the sub-population such as pregnant women. Was the debate pertaining to the Japanese population or the populations in the West? Would this make any difference? Moreover, did the authors argue or imply that the Patient Health Questionnaire-9 (PHQ-9) works better than the DSM? If so how and why? Or simply because the instrument was included in their study? This type of claim must be well-guarded!

We added the following sentences at the end of Limitation.

Of course, since our samples are all Japanese women, we should be sensitive to cultural or linguistic differences. Cross cultural studies should be performed colleting poplations from multiple countries. This is, however, beyond the scope of the current study, and awaits future international studies.

As to the difference between PHQ-9 and DSM, we should like to once more emphasize that the PHQ-9 is a questionnaire tool to estimate (presume) DSM’s major depressive episode. We used the PHQ-9 because our study was a questionnaire survey. If this had been an interview survey, we would have used the SCID.

 Only a small portion of the pregnant women participated in the study. Any possible selection biases?

The reviewer was correct that there was possible selection bias. Nevertheless, the participation rate of this range is inevitable in epidemiological studies (at least in Japan) and we have no means to identify ‘selection’ bias.

 Please report or indicate the goodness-of-fit of the GRM model(s). Do these models fit the data?

We are unaware of means to measure ‘goodness-of-fit’ of IRT.